



# Precipitation and Microphysical Processes Observed by

# Three Polarimetric X-Band Radars during HOPE

Xinxin Xie[1], Raquel Evaristo[1], Clemens Simmer[1], Jan Handwerker[2] and Silke

Trömel[1],

[1]Meteorological Institute, University of Bonn, Bonn, Germany

[2]Institute of Meteorology and Climate Research , Karlsruhe Institute of Technology,

Karlsruhe, Germany

*Correspondence to*: Xinxin Xie (xxie@uni-bonn.de)

**Abstract.** This study presents a first analysis of precipitation and related

microphysical processes observed by three polarimetric X-band Doppler radars

(BoXPol, JuXPol and KiXPol) in conjunction with a ground-based network of

disdrometers, rain gauges and vertically pointing micro rain radars (MRR) during the

High Definition Clouds and Precipitation for advancing Climate Prediction $(HD(CP)^2)$

Observational Prototype Experiment (HOPE) during April and May 2013 in Germany.

While JuXPol and KiXPol were continuously observing the central HOPE area near

Forschungszentrum Juelich at a close distance, BoXPol observed the area from a

distance of about 48.5 km. MRRs were deployed in the central HOPE area and one

MRR close to BoXPol in Bonn, Germany. Seven disdrometers and three rain gauges

providing point precipitation observations were deployed at five locations within a 5

$\times 5$ $km^2$ region, while another three disdrometers were collocated with the MRR in

Bonn. The daily rainfall accumulation at each rain gauge/disdrometer location

estimated from the three X-band polarimetric radar observations showed a very good

agreement. Accompanying microphysical processes during the evolution of

precipitation systems were well captured by the polarimetric X-band radars and

corroborated by independent observations from the other ground-based instruments.





# 1. Introduction

In the frame of the project "High Definition Clouds and Precipitation for advancing Climate Prediction" $(HD(CP)^2)$, which aims at improving the accuracy of climate models in relation to cloud and precipitation processes, the $HD(CP)^2$ Observational Prototype Experiment (HOPE) was conducted during April and May 2013 within the study area of the Transregional Collaborative Research Center 32 (Simmer et al., 2015) in the vicinity of the Juelich ObservatorY for Cloud Evolution (JOYCE) in Germany (Löhnert et al., 2015). The HOPE was conducted in order to provide observations for high-resolution climate models and to improve our understandings of cloud and precipitation processes.

An array of ground-based instruments deployed during HOPE provided comprehensive cloud and precipitation process observations. In this study we concentrate on the precipitation monitoring instruments. Three polarimetric X-band Doppler radars installed in Bonn (BoXPol) and in the vicinity of the JOYCE site (JuXPol and KiXPol), respectively, were operated together to continuously monitor 3D precipitation patterns in order to obtain a holistic view of precipitating systems from micro- and macro-physical perspectives. BoXPol and JuXPol were installed at a distance of 48.5 km from each other and were operated by the Meteorological Institute of the University of Bonn and the TERENO program of the Helmholz Association (http://teodoor.icg.kfa-juelich.de, Zacharias et al., 2011), respectively (see Diederich et al., 2015a for details on both radars), while KiXPol, which was ~9.6 km (~50.6 km) away from JuXPol (BoXPol), was deployed by the Karlsruhe Institute of Technology (KIT). A network composed of rain gauges and disdrometers measured local precipitation, and collocated Micro Rain Radars (MRR) simultaneously measured vertical profiles of precipitation and raindrop size distributions (DSD).

Dual-polarization radars provide multiparameter measurements, which improve quantitative precipitation estimation (QPE) compared to single polarization radars (Zrnic and Ryzhkov, 1999; Zhang et al., 2001; Brandes et al., 2002; Ryzhkov et al.,



2014). A thorough comparison of retrieval algorithms for rainfall estimation using
polarimetric observables for the HOPE area can be found e.g. in Ryzhkov et al. (2014)
and Diederich et al. (2015b). Many studies has already shown the potential of
polarimetric radars to identify fingerprints of macro- and micro- physical processes
related to the evolution of precipitation systems (Kumjian and Ryzhkov, 2010, 2012;
Kumjian et al., 2012; Andric et al., 2013; Kumjian and Prat, 2014), based on the
sensitivities of polarimetric observables to particle size, shape, concentration and
composition (Bechini et al., 2013; Ryzhkov and Zrnic, 1998; Giangrande et al., 2008).
E.g., very few large rain drops near the ground or at the leading edge of a rain cell
result in a larger mean particle size and induce strong differential reflectivity ($Z_{DR}$)
accompanied by small reflectivity (Z), which indicates the occurrence of size sorting
(Kumjian and Ryzhkov, 2012). Increasing mean particle sizes due to evaporation and
coalescence may enhance $Z_{DR}$, while Z is reduced during evaporation by the depletion
of small rain drops (Kumjian and Ryzhkov, 2010; Li and Srivastava, 2001). Z, $Z_{DR}$
and specific differential phase ($K_{DP}$) all decrease when large raindrops break up
(Kumjian and Prat, 2014). Such information thus can be used to validate cloud and
precipitation parameterization schemes.
The paper is structured as follows. Section 2 introduces the instrumentation deployed
during HOPE, while Section 3 presents the surface rainfall estimated from the radars,
in conjunction with disdrometers and rain gauges. Section 4 presents and discusses
the development of different precipitation systems and related microphysical
processes. Size sorting due to vertical wind shear and coalescence will be illustrated
via the combination of two X-band polarimetric radars. Another case of size sorting
captured by BoXPol and a nearby MRR and disdrometers will also be examined in
detail. Finally, observed riming/aggregation signatures will be discussed. Conclusions
will be given in Section 5.
**2. Instrumentation**





## 2.1 Three X-band polarimetric radars

The three polarimetric X-band Doppler radars BoXPol, JuXPol, and KiXPol were
operating at a frequency of 9.375 GHz. Topography and the locations of the radars,
disdrometers, rain gauges and MRRs are shown in Fig. 1. While JuXPol and KiXPol
were both performing observations in the vicinity of Juelich, Germany, BoXPol
observed the HOPE area from a distance of about 48.5 km on the roof of a building
next to the Meteorological Institute of the University of Bonn in Bonn, Germany,
collocated with one OTT Parsivel and two Thies optical laser disdrometers. The three
polarimetric radars provide the standard polarimetric variables observed in a
simultaneous transmit and receive (STAR) mode, namely $Z$, $Z_{DR}$, $K_{DP}$, and $\rho_{HV}$
(copolar correlation coefficient) in addition to the radial Doppler winds and its
variance. Detailed technical specifications of JuXPol and BoXPol can be found in
Diederich et al. (2015a) and for KiXPol under www.imk-tro.kit.edu/english/5438.php.
The calibration bias of the three radars were corrected following Diederich et al.
(2015a).
Figure 2 shows the operation duration of the three polarimetric radars during HOPE.
BoXPol had technical problems on 15 May 2013 and was back to work at around
0800UTC on 16 May 2013. JuXPol performed observation from 5 to 8 April 2013.
Afterwards, no measurements were available until 22 April 2013 due to technical
problems. From 26 to 29 April 2013, JuXPol was only taking range height indicators
(RHI) at 233.7° azimuth oriented towards JOYCE every minute. KiXPol started its
observations on 3 Apr 2013 but had two breakdowns during April. In May, when
KiXPol was performing only RHI scans on request, no PPIs were available.





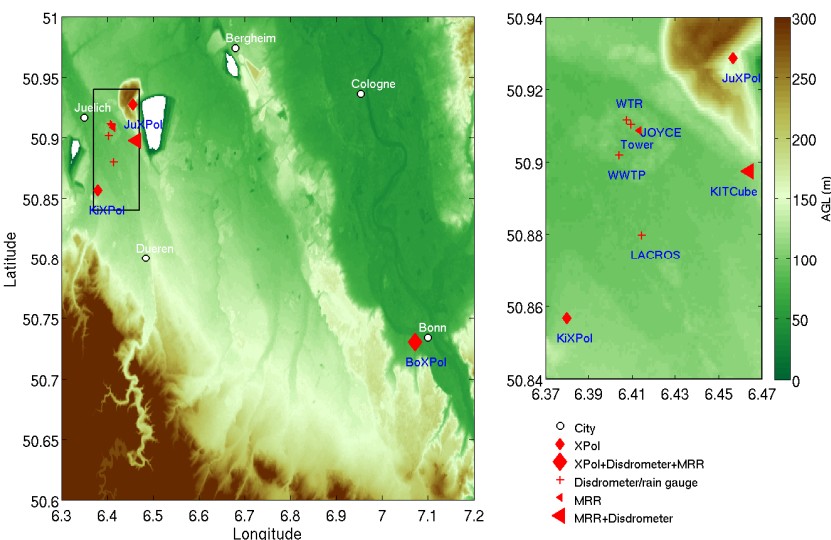

Figure 1. Location of the three polarimetric X-band radars (XPol) and associated
micro rain radars (MRR), rain gauges and disdrometers during HOPE. The right panel
is the zoomed-in region of the black box area on the left. The red diamond markers
indicate the locations of the X-band polarimetric radars, the red crosses indicate the
locations of disdrometers and/or rain gauges at the sites of LACROS (the Leipzig
Aerosol and Cloud Remote Observations System), KITCube (Kalthoff et al., 2013),
WWTP (wastewater treatment plant), Tower, WTR (wind-temperature-radar), and the
red triangles are the MRR locations at JOYCE and KITCube. White areas (elevations
below sea level) are open-pit mines.
The three polarimetric X-band radars were performing volume scans consisting of
stacked plan position indicators (PPI) with different scan strategies (Table 1). In
addition to the volume scans, BoXPol and JuXPol also performed RHIs and vertical
scans. A full volume scan of BoXPol and JuXPol takes about 5 min; in between RHI
scans and one vertical scan (bird bath scan) were performed. The two RHIs of
BoXPol were oriented towards JOYCE (290°) and LACROS (293.4°) after 9 April
2013, while JuXPol made RHIs only towards JOYCE. JuXPol made RHIs every



minute between 26 and 29 April 2013 followed by volume scans with PPIs at 10
elevations and one RHI and vertical scan in 5 minute intervals. KiXPol performed
only volume scans at 14 elevations every 5 minutes from April 2013 on (see Table 1).
In May 2013, volume scans were interrupted on demand and instead RHI scans
directed towards the prevailing wind direction were performed with a temporal
resolution of 1 minute (Fig. 2).

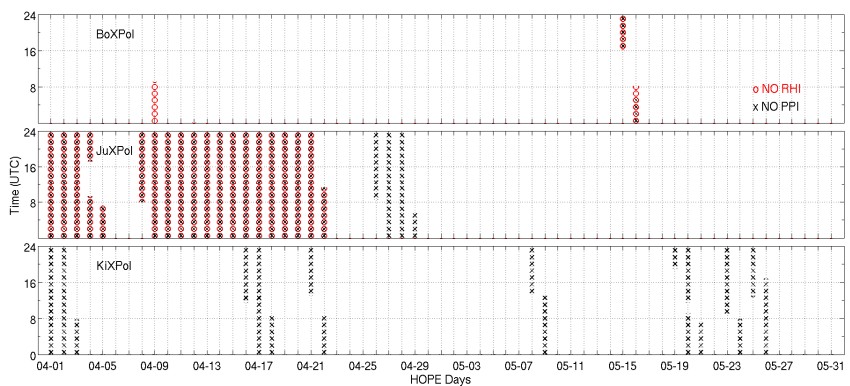

Figure 2.    Operation time of the polarimetric X-band radars BoXPol, JuxPol, and
KiXPol during HOPE from 1 April to 31 May 2013, with the red circles indicating
"no range height indicator (RHI) available" and the black crosses indicating "no plan
position indicator (PPI) available". KiXPol performed only RHIs on demand in May
where "no PPI" was marked. The general scan strategies of the three polarimetric
X-band radars are described in Table 1.



Table 1   The three polarimetric radars during HOPE

|  | BoXPol | JuXPol | KiXPol |
|---|---|---|---|
| location in latitude/longitude | 50.73°/7.07 ° | 50.93 °/6.46 ° | 50.86 °/6.38 ° |
| elevation in m a.s.l. | 100.0 | 310.0 | 116.6 |
| PPIs at elevation in ° | 1/1.5/2.4/4.5/7/8.2/ 11/14/18/28 | 1/2/3.1/4.5/6/8.2/ 11/14/18/28 | 0.6/1.4/2.4/3.5/4.8 /6.3/8/9.9/12.2/14.8/ 17.9/21.3/25.4/30 |
| RHIs at azimuth in ° | 309.5/298.6 (1-8 April); 290.0/293.4 (9 April-31 May) | 118.6 (1-25 April); 233.7 (26 April-31 May) | on request in May |
| bird-bath scan | yes | yes | No |
| radial resolution in m | 100 - 150 | 100 - 150 | 250 |
| scan period | every 5 min | every 5 min | every 5 min |

## 2.2   Rain gauges, disdrometers, MRRs and radiosondes

In the vicinity of JOYCE, disdrometers and rain gauges were installed within an area
of approximately 25 km$^2$. Seven disdrometers observed surface rain rates and DSDs
while three rain gauges measured rain accumulations (Table 2). The disdrometers and
rain gauges close to Juelich are used to evaluate radar derived QPE. Disdrometer
observations at BoXPol which is ~48.5 km away from JuXPol are not taken into
account, considering the spatial and temporal variability of rainfall.
Three MRRs were deployed close to JOYCE, KITCube and BoXPol. At JOYCE and
KITCube, the MRRs measured vertical DSD profiles with a vertical resolution of 100





m, at BoXPol 150 m. Due to the near field scattering effects, MRR observations at the
first 3 gates are not used.
Radiosondes were launched regularly twice per day close to KITCube, one at 1100
UTC and another at 2300 UTC. Additional radiosondes were launched during
intensive observation periods (IOPs).
Table 2    Information on rain gauges and disdrometers deployed during HOPE

| Site name | Location in (Latitude, Longitude) | Instrument (quantity) | Temporal resolution (s) | operation period |
|---|---|---|---|---|
| KITCube | (50.90°,6.46°) | Joss-Waldvogel disdrometer (1) | 60 | 1 Apr - 31 May 2013 |
| | | OTT Parsivel2 (1) | 60 | 1 Apr - 31 May 2013 |
| LACROS | (50.88°,6.41°) | OTT Parsivel2 (1) | 30 | 2 May - 31 May 2013 |
| WTR | (50.91°,6.41°) | OTT Parsivel2 (1) | 30 | 17 Apr - 31 May 2013 |
| | | OTT Pluvio (1) | 10 | 17 Apr - 31 May 2013 |
| WWTP | (50.90°,6.40°) | OTT Parsivel2 (1) | 30 | 17 Apr - 31 May 2013 |
| | | Tipping bucket rain gauge (1) | -- | 17 Apr - 31 May 2013 |
| Tower | (50.91°,6.41°) | OTT Parsivel1 (1) | 30 | 17 Apr - 31 May 2013 |
| | | OTT Parsivel2 (1) | 30 | 17 Apr - 31 May 2013 |
| | | OTT Pluvio (1) | 10 | 17 Apr - 31 May 2013 |
| BoXPol | (50.73°,7.07°) | OTT Parsivel2 (1) | 30 | 1 Apr – 31 May 2013 |
| | | Thies Disdrometer (2) | 60 | 1 Apr – 31 May 2013 |
| | | OTT Pluvio (1) | 60 | 1 Apr – 31 May 2013 |



## 3. Precipitation during HOPE

We first compare QPE derived from the polarimetric radar observations with the observations of the surface network of rain gauges and disdrometers, in order to corroborate the consistency and accuracy of both estimates.

Figure 3 shows the daily rain accumulation and precipitation duration averaged over the rain gauge/disdrometer observation sites in the HOPE region (Fig. 1). For rainfall duration, only disdrometer observations are used since the weighing-type rain gauges often indicate small noisy rain-like signals, which prevent accurate information on rainfall duration. According to these observations, the maximum daily rain accumulation was ~14.5 mm, the total rain accumulation during HOPE was ~104.8 mm, and the total rainfall time was ~144 hours, i.e., 10% of the total HOPE period. The rainfall observations at the five locations are in good agreement with each other, as indicated by the bars in Fig. 3, which show the full range of the observations.

According to the disdrometer observations, precipitation during HOPE was not very intense (Fig. 4). The distribution of rain intensities was calculated based on individual measurements of disdrometers, instead of averaging over disdrometer sites at a single time step. Rain rates determined at a temporal resolution of 1 minute were below less than 2 mm h$^{-1}$ for more than 88% of the total precipitation duration, while rain rates above 5 mm h$^{-1}$ were observed for less than 3 hours. Only one hour of rain rates above 8 mm h$^{-1}$ did occur.



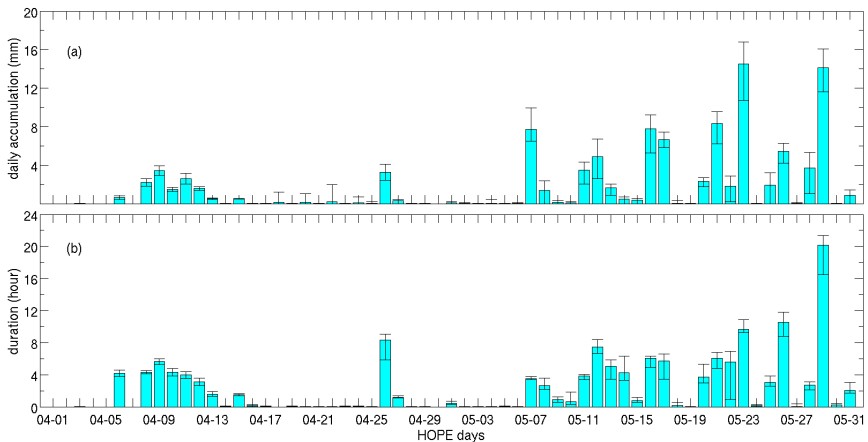

Figure 3.   (a): Daily rainfall accumulation during HOPE. The height of the columns
indicates the mean value while the bars indicate the range of the maximum and
minimum rain accumulations observed by the 3 rain gauges and 7 disdrometers at the
five station locations (Fig. 1). (b): Daily precipitation duration derived only from the 7
disdrometers (see discussion in the text). Again the bars denote the range of the
observations.

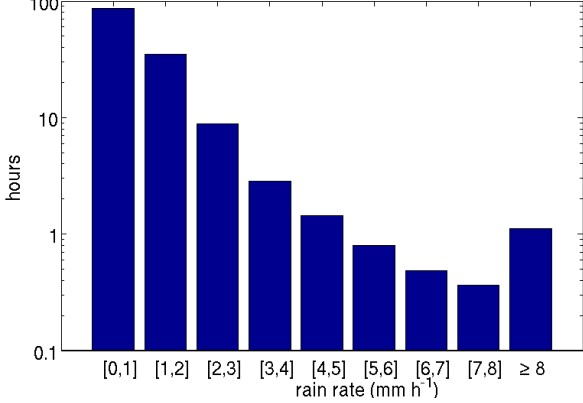

Figure 4.   Distribution of rain intensities observed over one minute by the
disdrometers in the inner HOPE area.





In accordance with the relatively light rainfall events during HOPE, the polarimetric
radar observables $Z_{DR}$ and $K_{DP}$ were low and quite noisy. Under these conditions,
most of the time, we simply used Marshall-Palmer relation for quantitative rainfall
estimations (Marshall and Palmer, 1948),
$$Z_H = 200R^{1.5} \quad (or \quad R = 0.029Z^{0.67}) \qquad (1)$$
where $Z_H$ (in mm$^6$ m$^{-3}$) is the radar reflectivity for horizontal polarization in linear
scale and $R$ is the rain rate in mm h$^{-1}$.
Since Equation (1) tends to overestimate stronger rain intensities (Zrnić et al., 2000;
Trömel et al., 2014b), the R-$K_{DP}$ relation is employed for rain rate estimation when $Z_H$
is above 37 dBz, i.e., the instantaneous rain rate is above 8 mm h$^{-1}$ (Diederich et al.,
2015b; Ryzhkov et al., 2014). $K_{DP}$ is independent of calibration and unaffected by
attenuation (Ryzhkov et al., 2014). Thus, following Diederich et al. (2015b) and
Ryzhkov et al. (2014), in this case the rain rate is determined by
$$R = 16.9K_{DP}^{0.801} \qquad if \ K_{DP} > 0 \qquad (2)$$
where $K_{DP}$ is the specific differential phase (° km$^{-1}$) and filtered from polarimetric
radar measurements following Hubbert and Bringi (1995).
Radar bins with copolar correlation coefficient $\rho_{HV} < 0.75$ have been neglected in
order to eliminate the ground clutter contamination. For JuXPol and KiXPol,
observations at elevations 4.5° and 3.5°, respectively, are used to calculate the rain
rates and avoid the possible impacts from a 120-m height meteorological tower at
Forschungszentrum Juelich, while an elevation of 1° is chosen for BoXPol rainfall
estimation since the radar beam at longer distance is less affected by the ground
clutter and certainly overshoots the meteorological tower. The mean beam diameter of
BoXPol over the HOPE area is around 850 m, which is almost 10 times larger than
that of JuXPol and KiXPol, and its beam height (~860 m) is about 2 times larger
comparing to JuXPol and KiXPol.





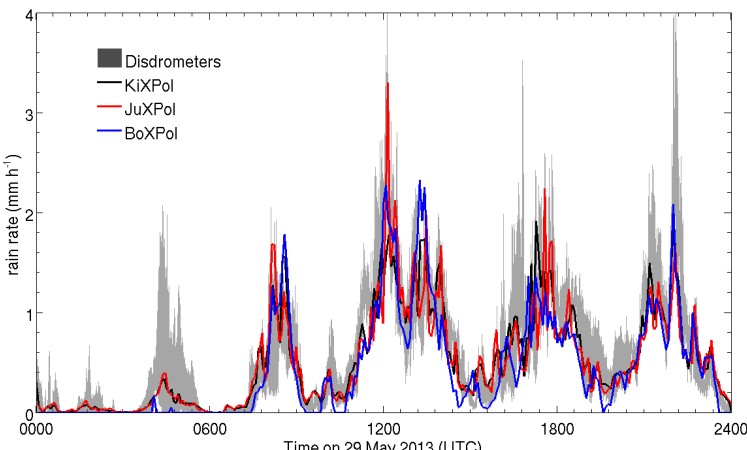

Figure 5.    Time series of rain rates derived from observations of the seven
disdrometers and the three polarimetric radars on 29 May 2013. The shaded gray area
indicates the range of rain rates observed by the disdrometers with 1-min temporal
resolution in the HOPE area while the rain rate from the three polarimetric radar
observations is calculated at the radar gates that are coincident with disdrometer
locations and also averaged over the five disdrometer locations.
Figure 5 compares as an example the mean rain rates derived from the three X-band
polarimetric radar over the five disdrometer locations with the disdrometer
observations for 29 May 2013. Precipitation fell intermittently with five more intense
periods separated by short periods of no or very low rain rates and maximum rain
rates between 1 and 3 mm h$^{-1}$. In general, the variability of the radar-derived surface
precipitation matches very well the disdrometer measurements. JuXPol and KiXPol
are in a better agreement with the surface measurements than BoXPol for the very
lower rain rates, which probably suffers from the effects of non-uniform beam filling
effects due to the much larger distance from the HOPE area (Giangrande and Ryzhkov,

18   2008).





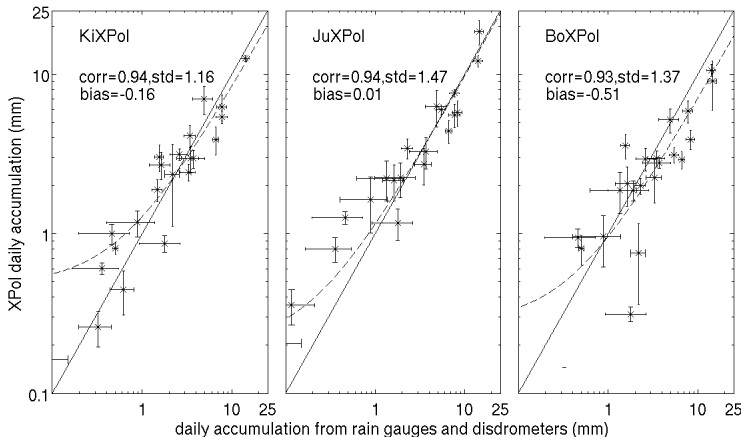

Figure 6.    Mean daily radar-derived rain accumulation over the disdrometer/rain
gauge locations, compared to the surface precipitation observed by the rain gauges
and disdrometers in the HOPE area. The bars indicate the standard deviation of the
estimates from the particular radar (vertical bars) and from the surface observations
(horizontal bars) . The dashed black line is the best linear fit of the daily rain
accumulation on the logarithmic scale while the solid black line is the 1:1 line.
Daily-accumulated rainfall estimated by the three polarmetric radars are compared
with the observations of rain gauges and disdrometers in Fig. 6. Both estimates are
very consistent as indicated by correlations above 0.93. As for 29 May 2013, BoXPol
estimates result in lower daily accumulations than for the other two radars, again
probably caused by beam broadening (Giangrande and Ryzhkov, 2008).





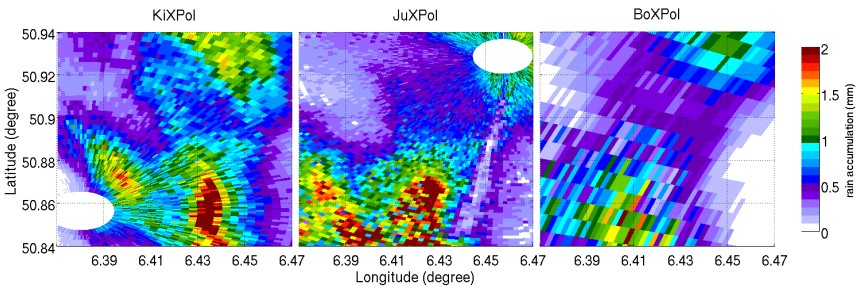

Figure 7. Rain accumulation over the HOPE area between 0830 and 0900 UTC (6
PPIs) on 29 May 2013 observed by the three polarimetric radars.
With a range resolution of 150/250 m and a beam diameter of approximately 87/850
m over the HOPE area, the three polarimetric radars allow to characterize the
precipitation patterns in the HOPE domain in high resolution, which will be important
for model evaluation. A 30-min rain accumulation over the inner HOPE area on 29
May 2013 shows that, the three radars estimates result in quite similar precipitation
patterns. Bins close to KiXPol and JuXPol were contaminated by ground clutters
while the beam broadening and height at the larger ranges deteriorates the similarity
between the BoXPol and KiXPol/JuXPol estimates (Fig. 7). A combination of the
three radar observations will definitely be an advantage to reconstruct the
precipitation patterns over the HOPE area. Since no adjustments of the R-$Z_H$ and
R-$K_{DP}$ relations were made, these results are very promising. The three radar
estimates together with the direct comparisons with the rain gauges and disdrometers
allow to attribute robust error estimates to these precipitation fields, which will be
very valuable when compared with model simulations.
**4. Observed microphysical processes**
Falling hydrometeors are subject to growth and/or depletion by a range of
microphysical processes which leave their fingerprints in the spatial and temporal





evolution of several polarimetric moments. Since microphysical processes are
simulated in atmospheric models with increasing details, polarimetric radar
observations can be used for model validations and thus spur further improvements.
In this section we present three cases, where such microphysical processes could be
observed by the radars and substantiated by MRR and disdrometer observations.

## 6 4.1 Case 1: Size sorting and coalescence

On 26 Apr 2013, a cold front passed over Germany, which came with a large band of
stratiform rain that persisted from the morning hours until the end of the day. The
daily rain accumulation recorded by the surface observations was about 3.5 mm while
the precipitation lasted up to 8 hours (Fig. 3). Six radiosondes launched at KITCube at
0700 UTC, 0900 UTC, 1100 UTC, 1300UTC, 1600 UTC and 2300 UTC, respectively,
recorded a freezing level above 2100 m during daytime, which descended down to
830 m at about 2300 UTC.

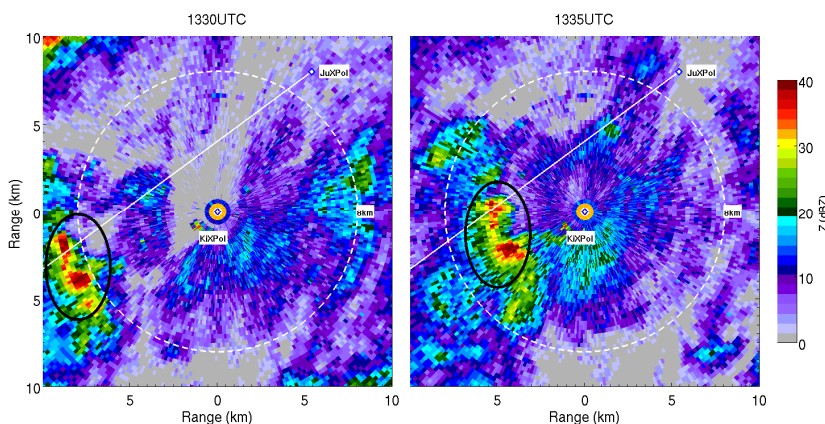

Figure 8. Reflectivity ($Z_H$) of KiXPol observed at an elevation angle of 3.5° at 1330
UTC and 1335 UTC on 26 April 2013. The precipitating cell examined in the text is
highlighted by the black ellipse. The white solid line indicates the azimuth direction
of the JuXPol RHIs, while the white dashed circle delineates the 8-km distance from
KiXPol.





KiXPol preformed volume scans every 5 min on that day, with scan elevations
ranging from 0.6° to 30° (Table 1), while JuXPol made RHI scans in the direction of
JOYCE every minute.

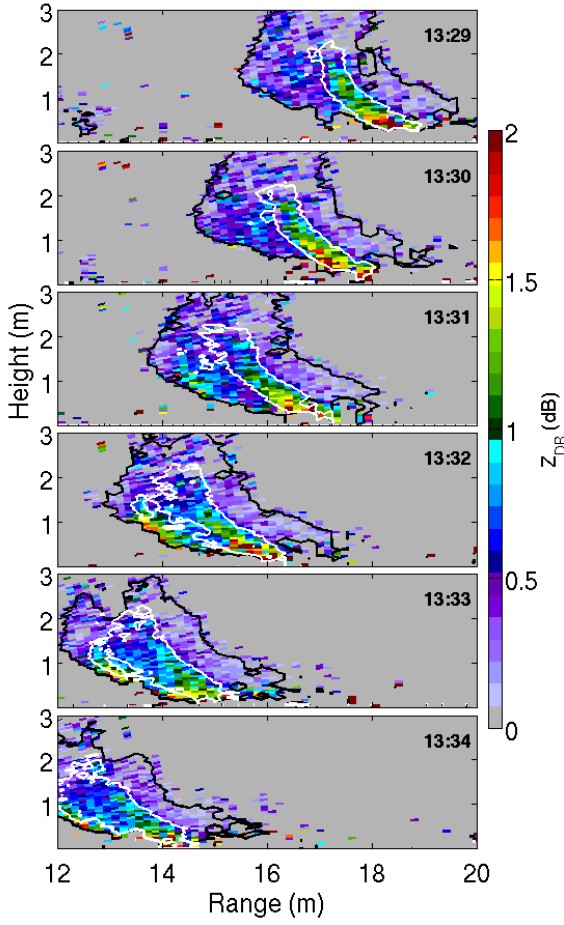

Figure 9.   Sequence of RHIs of differential reflectivity ($Z_{DR}$) measured by JuXPol at
an azimuth angle of 233.7° between 1329 UTC and 1334 UTC on 26 April 2013
(from top to bottom). The contour lines indicate reflectivity values ($Z_H$) of 15 dBz
(black) and 30 dBz (white), respectively.




At 1330 UTC KiXPol observed a precipitating cell approaching the radar from the
southwest at about 10 km distance, which was moving towards JuXPol (Fig. 8). At
1335 UTC the cell was within 8 km from KiXPol, where it started to dissolve (not
shown). RHIs performed with JuXPol at the azimuth direction 233.7° nicely tracked
the approaching cell (Fig. 9).
The high temporal resolution of the JuXPol RHIs allows for a detailed insight into the
evolution of the precipitating cell. The cell was first observed by JuXPol at 1300 UTC
at about 45 km distance and kept moving towards JuXPol with low reflectivities at
about 20 dB (not shown). At 1329 UTC, JuXPol detected the precipitating cell
entering its RHI at 20 km range (Fig. 9). In the center of the precipitating cell tilted
towards the northeast by the wind shear (See Fig. 10), near surface $Z_{DR}$ values were
up to 2 dB while $Z_H$ was above 30 dBz. $Z_{DR}$ increases towards the ground concurrent
with an increasing $Z_H$. This behavior is a clear sign of coalescence, which shifts small
raindrops to larger sizes and increases the mean raindrop size (Kumjian and Prat,

15   2014).

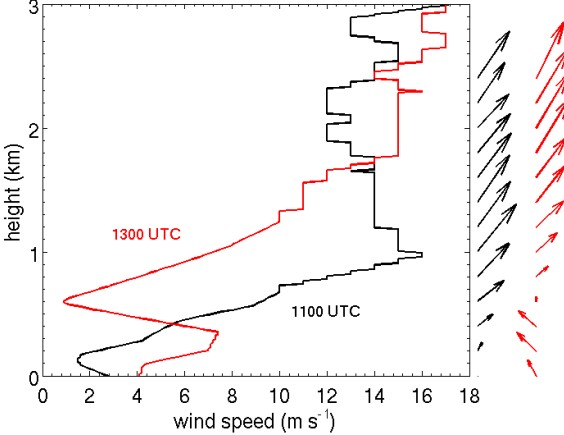

Figure 10 Wind profiles derived from radiosondes launched at KITCube at 1100 UTC
and 1300 UTC. The arrows on the right indicate the wind vector (0° indicates the
north) while their lengths are proportional to wind speed.



While moving towards JuXPol, the tilt of the cell led to a concentration of large rain
drops at the leading edge of the precipitating cell, where their larger fall speed
separates them from the smaller droplets which largely remain in the flow volume
(e.g., Kumjian and Ryzhkov (2012)). From 1329 UTC to 1330 UTC, $Z_{DR}$ at the
leading edge of the cell is below 0.5 dB (Fig. 9). At 1331 UTC, $Z_{DR}$ begins increasing
and later on reaches up to 2 dB while $Z_H$ remains in the order of 15 dBz in that region.
When the cell begins to dissipate as it moves forward, $Z_{DR}$ decreases down to ~ 1 dB
both in the center and upstream of the precipitating cell.

### 4.2  Case 2: Size sorting due to vertical wind shear

A second case on size sorting caused by the vertical wind shear was well captured by
BoXPol on 17 May 2013. A deep low pressure system reaching from the surface up to
200 hPa was found over the Northeast Atlantic and the British Isles on the previous
day, while a surface low was moving from the western Mediterranean to the north,
towards central Europe. As a result a complex pattern of fronts was affecting France
and Germany due to the interaction of both systems. On 17 May 2013, a stationary
front along with a through of warm air aloft passed over West Germany, moving
eastwards. Low atmosphere levels were characterized by high humidity and a sharp
West-East temperature gradient. A band of mostly stratiform rain affected
south-western and western Germany earlier in the day, while later on convective rain
with lightning activity developed over south east and central Germany. About 8 mm
of rain accumulated over 6 hour time spans as recorded by the disdrometers (Fig. 3).





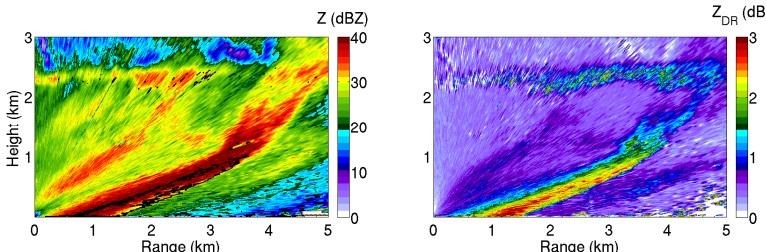

Figure 11.    Reflectivity ($Z_H$, left) and differential reflectivity ($Z_{DR}$, right) observed by
BoXPol at an azimuth angle of 290° at 1240 UTC on 17 May 2013. The black isoline
in the left panel indicates the 2-dB $Z_{DR}$ contour line.
The melting layer can be easily identified by the enhanced $Z_H$ and $Z_{DR}$ at an altitude
of ~2.2 km in the BoXPol RHI performed at an azimuth angle of 290° (Fig. 11).
Similar to the first case presented above, the strong $Z_{DR}$ at the leading edge indicates
the increase of mean raindrop size due to the accumulation of large raindrops by size
sorting.




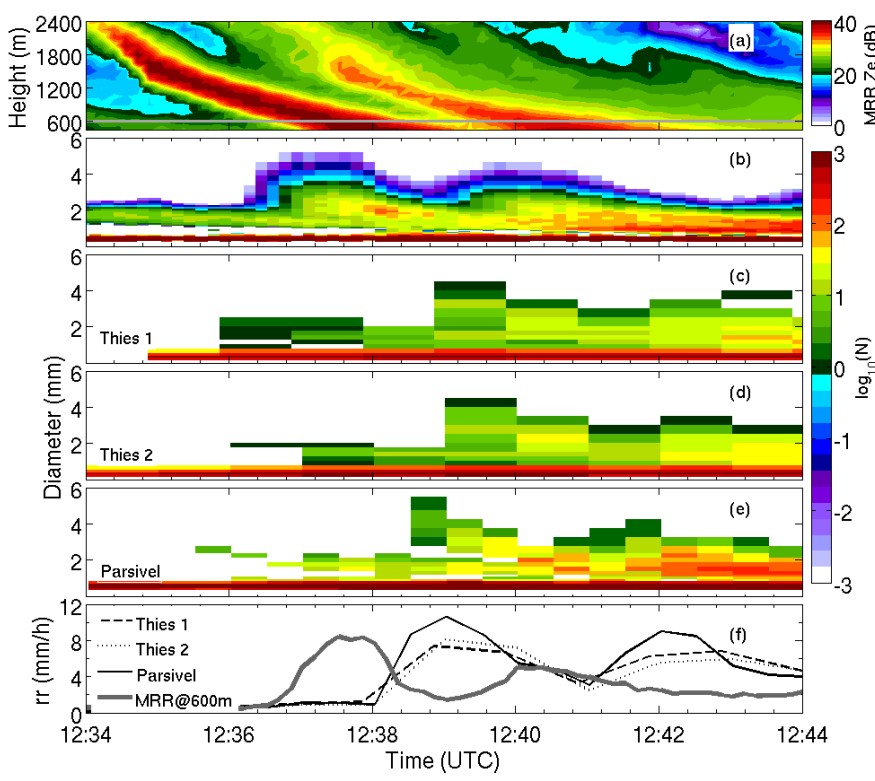

Figure 12. Different instrument observations located within distances of 5 meters

close (200 m) to the BoXPol location in Bonn, Germany, between 1234 UTC and

1244 UTC on 17 May 2013. (a): Reflectivity observed by vertically pointing micro

rain radar (MRR). The grey horizontal solid line indicates the 600 m height level. (b):

MRR-observed DSDs at 600 m altitude. (c) DSDs observed by a Thies disdrometer

with its transmitter and receiver line pointing along the east-west direction (Thies 1);

(d) same as (c) but for a Thies disdrometer pointing along the south-north direction

(Thies 2); (e) Same as (c) except for an OTT Parsivel disdrometer; (f) Rain rate

observed by an MRR at 600 m height and the three disdrometers collocated with the

MRR at the BoXPol station.





At 200 m distance from BoXPol, vertical profiles of DSDs were observed by an MRR.
Figure 12 shows the time series of MRR-derived reflectivity (Panel a) with the
corresponding DSDs at an altitude of 600 m (Panel b). The first cell of a precipitation
system passed BoXPol and the MRR before 1240 UTC with reflectivities up to 40
dBz in the center, followed by a second peak with reflectivities up to 35 dBz (Fig.
12a). The derived DSDs indicate that, fast falling large raindrops tend to concentrate
at the upstream side of the cell, while raindrops less than 3 mm in diameter have a
larger number concentration downstream (Fig. 12b).
The OTT Parsivel and Thies optical laser disdrometers collocated with the MRR also
captured the precipitation event on that day (Fig. 12c-12f). One Thies disdrometer
was deployed with its transmitter-receiver line in the west-east direction (Thies 1) and
the other in the south-north direction (Thies 2). For the surface DSDs shown in Fig.
12b-12e, the largest raindrops collected by the two Thies disdrometers are below 4
mm after 1239 UTC. Similar to MRR observations, however, the Parsivel observed
larger raindrops up to 5 mm at an earlier time step since it was operated at a temporal
resolution of 30 s. It implies that a temporal resolution of better than 1 min is required
to better interpret the DSD evolution caused by size sorting due to vertical wind shear
and to improve the surface rainfall estimations.
The surface rain rates observed by the three disdrometers differ from the MRR
observations at 600 m considering the spatial and temporal shifts (approximately 2
min) (Fig. 12f). The maximum rain rate estimated from the MRR at 600 m is ~ 8 mm
h$^{-1}$ at 1238 UTC, with a second peak of ~ 6 mm h$^{-1}$ at 1240 UTC. While the Parsivel
overestimates the rain rates, comparing to the MRR observations, the two Thies
disdrometers tend to slightly underestimate the rain rates with their rough time
resolution (1 min). Nevertheless, these observations are consistent with the occurrence
of the size sorting process shown from the radar observations..
**4.3  Case 3: Riming/aggregation processes observed by JuXPol**
On 29 May 2013 a cut-off process was underway over western and middle Europe,





resulting in a broad and well defined upper level vortex. At lower levels the pressure
distribution was more complex with several small surface lows and generally weak
pressure gradients. One of these surface lows, initially situated over southern England
at 0000 UTC, moved to eastern France during the day. The corresponding cold front
became quasi-stationary, as indicated by a sharp $\theta_e$ (equivalent potential temperature)
gradient over Be-Ne-Lux and western Germany (not shown). At 0000 UTC and 0600
UTC frontogenetic forcing was strongest due to deformational processes in the
vicinity of the front as it interacted with a second low over the northern half over
Germany. This resulted in a subsequent reinforcement of frontal precipitation over the
HOPE area until 1200 UTC. During and after that intensification period the frontal
temperature gradient gradually dissolved due to evaporative cooling and the advection
of a colder maritime air mass also on the warm side of the front. As a consequence
frontal precipitation weakened by the end of the day.
The daily rain accumulation for 29 May 2013 recorded by the surface observations
was ~14 mm while precipitation lasted up to 20 hours (Fig. 3): this was the day with
the longest rainy period which also lead to the second largest daily rain accumulation
during HOPE. Three radiosondes were launched at the location of KITCube, one at
2300 UTC on 28 May and two at 1100 UTC and 2300 UTC on 29 May. According to
the soundings, the freezing level was located at ~2.2 km at 2300 UTC on 28 May
2013 and subsided down to ~1.7 km at 1100 UTC on 29 May 2013.
Figure 13 shows so-called Quasi-Vertical Profiles (QVPs) of $Z_H$, $Z_{DR}$, $\rho_{HV}$ and $K_{DP}$
based on JuXPol measurements at 18° elevation angle between 0600 and 1430 UTC.
QVPs were first used by Trömel et al. (2014a) to reliably estimate backscatter
differential phase δ and Ryzhkov et al. (2016) further expanded the QVP methodology
and demonstrated its multiple benefits. The QVPs of polarimetric variables are
obtained by azimuthal averaging of the radar data collected during conical PPI scans
at higher antenna elevation angles in order to reduce statistical errors of the variables
and assign their average vertical profiles to a conical volume in a time-height display.
QVPs are especially beneficial for monitoring the temporal evolution of



microphysical processes active on a larger scale.

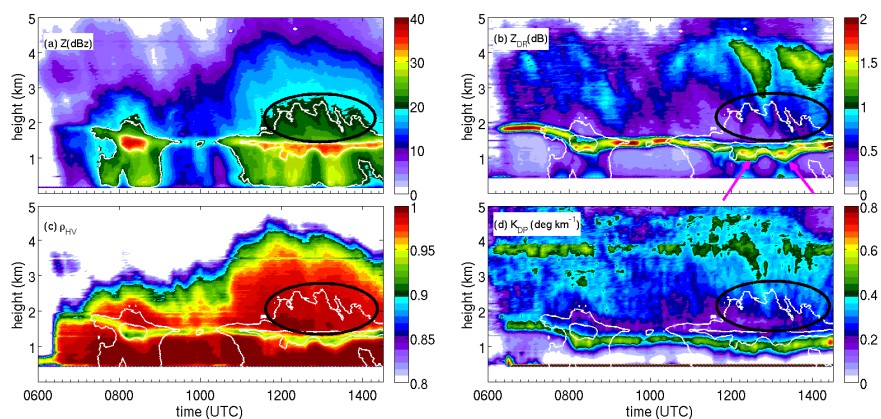

Figure 13. (a) Time series of Quasi-Vertical Profiles (QVPs) of $Z_H$ derived from
PPIs measured with JuXPol at 18° elevation on 29 May 2013 between 0600UTC and
1430UTC. The white lines indicate the 20 and 30 dBz contours of $Z_H$; (b), (c) and (d):
the same time series as (a) but for $Z_{DR}$, $\rho_{HV}$ and $K_{DP}$, respectively. The black ellipses
highlight the area for aggregation/riming while the magenta arrows in Panel (b)
indicate the $Z_{DR}$ saggings (see text for detail).
The most striking feature in Fig. 13 is the descent of the melting layer from 1.8 km
down to ~1.5 km height between 0700 and 0900 UTC. After 1200 UTC, a region
enhanced $K_{DP}$ above 3.5 km accompanied with $Z_{DR}>1.2$ dB aloft can be identified.
Bands of enhanced $Z_{DR}$ and bands of enhanced $K_{DP}$ are both considered as signatures
of dendritic growth (Kennedy and Rutledge, 2011). According to the radiosonde
ascending at 1100 UTC, the temperature zone of -10°C ~-15°C which favors the
growth of ice dendrites is located between 3.8 and 4.7 km. Thus, we may suspect
dendrites growing above 3.5 km especially after 1200 UTC (Fig. 13).
When following the height evolution of polarimetric variable structures above the
melting layer (ML) after 1200 UTC (Fig. 13), riming/aggregation processes are
indicated by enhancements of $Z_H$ and $\rho_{HV}$ above the ML while $Z_{DR}$ and $K_{DP}$ decrease
with height in unison above the ML after 1200 UTC (ellipses in Fig. 13). $Z_{DR}$ and $K_{DP}$





depressions aloft associated with increases in $Z_H$ and $\rho_{HV}$ above the ML suggest
increases of ice particle mean sizes due to riming and/or aggregation. Recently,
Moisseev et al. (2015) argued that the processes responsible for enhanced $K_{DP}$- and
$Z_{DR}$-bands might be different: they advocated that the $K_{DP}$ bands are caused by high
number concentrations of oblate relatively dense ice particles (early aggregates) and
are linked to the onset of aggregation processes, while $Z_{DR}$ bands in the absence of
$K_{DP}$ bands are observed when crystal growth is the dominating snow growth
mechansim and the number concentration is lower. Following their arguments, it can
also be speculated that aggregation processes are ongoing near the end of the
observation period shown in Fig. 13.
Discrimination between riming and aggregation is important for aviation security,
since riming implies the existence of supercooled liquid water above the freezing
level, which could result in dangerous icing on aircrafts. Riming is also associated
with embedded updrafts, convective development and thus precipitation enhancement.
In the presence of such updrafts, enhanced condensation of water vapor occurs and
leads to small liquid droplets which may be accreted by dry snowflakes. These rimed
snowflakes may grow fast and reach large sizes with higher terminal velocitiy before
they fall through the ML. Due to their enhanced terminal velocity, they melt at a
lower height and lead to the "sagging" signature of the bright band in terms of $Z_{DR}$
and $\rho_{HV}$ (Ryzhkov et al., 2016).
In Fig. 13, reduced $Z_{DR}$ combined with enhanced $Z_H$ and $\rho_{HV}$ above the ML occurs at
times, and also "sagging" signatures are clearly visible at around 1200 UTC and 1300
UTC (the magenta arrows in Fig. 13b). Starting from the bottom of the $Z_{DR}$- and $K_{DP}$-
bands at about 3 km height at 1200 UTC, $Z_{DR}$ decreases and Z increases downwards
most probably due to aggregation and/or riming. Here $Z_{DR}$ reduces down to a few
tenths of a dB just above the level where melting starts. However, this reduction is
expected to be more intense for riming than for aggregation. Riming makes the ice
particles more spherical leading to a lower $Z_{DR}$ by $0.1 - 0.3$ dB (Ryzhkov et al., 2016).
Thus, we speculate that riming causes the "sagging" effects of $Z_{DR}$ and $\rho_{HV}$ combined





with relatively low $Z_{DR}$ above the ML around 1200 UTC and 1300 UTC. To more
reliably distinguish between riming and aggregation, we require additional
measurements indicative e.g. of associated updrafts and supercooled liquid water
above ML, which could be provided by additional microwave radiometers and cloud
radars.
The discussed examples have clearly shown how polarimetric radars can be used to
identify and distinguish between different microphysical processes, like warm rain
processes and ice particle formation and growth. Converting the output of NWP
models into polarimetric radar variables and using a polarimetric forward radar
operator would provide an opportunity to validate the representation of the discussed
microphysical processes in such models.
## 5. Conclusions
This study presents a summary of rainfall observations and some examples of related
microphysical processes occurring during HOPE between 1 April and 31 May 2013.
At that time three X-band polarimetric Doppler radars observing the central HOPE
area of about 5 km $\times$ 5 km over which a surface network of rain gauges, disdrometers
and MRRs was deployed to assess the accuracy of the radar-based precipitation
observations and to demonstrate the capability of polarimetric radars to detect
microphysical processes. Rainfall accumulations at the daily and even hourly scale
were surprisingly consistent between the different observations demonstrating the
high quality of QPE based on R-Z and R-$K_{DP}$ relations at least for the low intensity
rainfall events prevalent during HOPE.
The combined observations of polarimetric radars and collocated instruments
demonstrated the ability of radar polarimetry to detect several microphysical
processes by so-called polarimetric fingerprints during the development and evolution
of precipitation systems. These fingerprints clearly identify microphysical processes
like coalescence, size sorting and riming/aggregation. Size sorting by wind shear was





e.g. well captured by the JuXPol and BoXPol RHI scans and corroborated by the
collocated MRR and disdrometer observations. While there were clear signs of other
processes like riming and aggregation, their distinction requires additional analysis in
conjunction with other independent observations e.g. from microwave radiometers,
lidars and cloud radars deployed at the JOYCE site, which is the focus of an ongoing
study.

## 8 Acknowledgements

This research was funded by the Federal Ministry of Education and Research in
Germany (BMBF) through the research programme "High Definition Clouds and
Precipitation for Climate Prediction - HD(CP)[2]" (FKZ: 01LK1219A and 01LK1210A).
We thank Martin Lennefer and Kai Muehlbauer at University of Bonn and the
SFB/TR 32 (Transregional Collaborative Research Center 32, http://www.tr32.de/)
funded by the German Research Foundation (DFG) for maintaining the instruments
and make the radar data available. We thank Patric Seifert from the Leibniz Institute
for Tropospheric Research (TROPOS) for providing the OTT Parsivel data at the
LACROS station. We also acknowledge Norbert Kalthoff and his colleagues from
the Karlsruhe Institute of Technology (KIT) for providing the observation data at
KITCube station and KiXPol observations during HOPE.

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
