# Peer review of "Three Polarimetric X-Band Radars and Ground-Based"

_Atmospheric Chemistry and Physics, 2016_

## Referee Comment (RC1) · Anonymous Referee #1 · 18 Mar 2016

GENERAL COMMENT:

This paper is well organized, in a clear and simple manner, starting by the description of the HOPE experiment and associated instrumental set up, and ending by an analysis of the three case studies from three polarimetric X-band radar observations.

The interest of the paper lies in taking advantage of multi-parameter measurement capability to improve or assess microphysical phenomena knowledge. The data interpretation is quite consistent and well referenced, and recent and interesting approaches are used (e.g. QVP by Ryzhkov et al., 2016). However, even if the paper aims at presenting some preliminarily step to some more ambitious study (as stated in the conclusion), the analysis shown would be worth being completed by additional measurements from the radars themselves. For instance, X-band Radar Doppler measurements in vertical pointing mode may alleviate some uncertainties about distinguishing between aggregation and riming above the melting layer (as mentioned in Case 3). At least, this could be mentioned or discussed in the paper.

Polarimetric measurements from the three radars are used in the microphysical analysis, as suggested by the title of the paper. Additionally, ground based instruments are used, not only to confirm, but also or to complete this analysis. The title should suggest the use of such complement.

SPECIFIC COMMENTS:

1°) Except for rain rate estimation, rain (or hail or melting snow) attenuation impacting X-band measurements is not mentioned at all in the paper. Indeed, the analyses use Z and ZDR, potentially biased by such attenuation. Are Z and ZDR corrected for attenuation?

2°) p7 lines7-9: Why are BoXPol disdrometer measurements not used, while MRR observations at the same site are?

3°) p9 lines 15-16: I do not understand how the distribution shown in Fig.4 results from individual measurements of disdrometers instead of averaging over disdrometer sites at a single time step. Does that mean, for instance, that 80 hours of [0,1] mm/h rain have been obtained by summing rainfall observation time over the N disdrometers (thus representing 80/N hours each)?

4°) p12 lines 16-17 and p13 line 13: the lack of consistency between rain rates also probably suffers from the representativeness error impacting BoXPol measurements (higher altitude of sampling volume).

5°) p14 Fig.7: When comparing KixPol and JuxPol rain accumulation, the south-west quarter of the panels shows significant differences. How could this be explained? Is this a problem of projection on ground?

6°) p25 line 3: About measurements indicative associate updraft, what about radar Doppler measurements at vertical incidence? (see general comment)

TECHNICAL CORRECTIONS:

1°) p1 line 21: replace "another three disdrometers" by "three other disdrometers".

2°) p14 line 9: remove "s" in radars (replace "three radars estimates" by "three radar estimates").

3°) p17 line 12: replace "30 dBz" by "30 dBZ".

4°) p23 line 11: add "of" after "a region".

---

## Referee Comment (RC2) · Anonymous Referee #2 · 11 Apr 2016

General Comments

This paper presents some data from an two months experiment. The examined dataset included three polarimetric X-band weather radars supplemented by MRRs, disdrometers and rain-gauges. The focus of the paper is on the radar observations. The recorded rain events were of low intensity and this didn't permit a more advanced evaluation of radars performance. Thus, instead of just showing some daily statistics and example data from three rain events with typical stratiform rain characteristics, the authors could present methods of data processing. For example, they have a network of three radars which overlap in the area of interest and, thus, a detailed comparison between the radars (and the rest of sensors like the MMR, disdrometers and raingauges)

could be performed. Furthermore, a method for construction of a mosaic with the quality controlled measurements from the three radars would be meaningful as a first data analysis. Also, the authors don't even mention the basic and critical processing algorithms of the radar data like the attenuation correction scheme and the handling of melting layer (bright band) effect on the estimated rain field.

Specific Comments

Section 2.1, Fig. 1: The setup of the systems shown in Fig. 1 is not optimal at all. Most of the systems (including two radars) are within 5 km distance. If this setup was intended for e.g. the study of small scale spatial distribution of rain this was shown in the paper.

p. 10, Fig. 4: The daily accumulated precipitation from the 7 disdrometers in Fig. 4b has larger range (minimum, maximum) compared to the range from the 3 rain gauges and the 7 disdrometers in Fig. 4a in some days (e.g. on 26 April), while obviously it should be less.

p. 14, lines 8-14: The conclusions of the authors about Fig. 7 are contradictory. First they say that precipitation patterns observed by the three radars, but immediately after the mention a lot of the many reasons why the observed patterns are different (which is the correct conclusion). They propose that a reconstruction of the precipitation pattern using a combination of all the radar data should be made, but as it was noted in the general comments they don't try to implement such a method.

p. 16, Fig. 9: There are not evident melting layer characteristics in the RHIs, even though it is mentioned in the text to move from 2100 m height down to 830 m during the event. It would be useful to include in Table 1 (or in a separate table) the operational parameters of the radar (like beamwidth, antenna rotation rate, sampling frequency etc.)

Section 4.2: In this section some data from MRR and disdrometers are shown. As

it was noted in the general comments the authors probably have enough data from the radars and these sensors to make a more detailed and useful comparison of their measurements. For example, a comparison of radar RHI data over (or near) the MRR site and MRR data would be an interesting comparison and study of the melting layer characteristics.

p. 21, lines 22-23: Why consider MRR data at 600m height as a reference (and not e.g. rain gauge data) and conclude that the Parsivels are overestimating rainfall rate? The MRR should be reduced to ground level using the time delay due to the average fall velocity of the droplets to have a proper comparison.

p. 23, Fig. 13: A comparison of QVPs and data from RHIs would be useful to understand the difference of QVP from actual vertical profiles and the limitations of this method.

---

## Author Comment (AC1) · 6 May 2016

**Response to Anonymous Referee #1**

**We thank the reviewer for his/her very valuable comments. We responded to the comments (in bold) and made modifications in the paper accordingly.**

**RC: reviewer comments**

**AR: author response**

GENERAL COMMENT:

RC1: This paper is well organized, in a clear and simple manner, starting by the description of the HOPE experiment and associated instrumental set up, and ending by an analysis of the three case studies from three polarimetric X-band radar observations.

The interest of the paper lies in taking advantage of multi-parameter measurement capability to improve or assess microphysical phenomena knowledge. The data interpretation is quite consistent and well referenced, and recent and interesting approaches are used (e.g. QVP by Ryzhkov et al., 2016). However, even if the paper aims at presenting some preliminarily step to some more ambitious study (as stated in the conclusion), the analysis shown would be worth being completed by additional measurements from the radars themselves. For instance, X-band Radar Doppler measurements in vertical pointing mode may alleviate some uncertainties about distinguishing between aggregation and riming above the melting layer (as mentioned in Case 3). At least, this could be mentioned or discussed in the paper.

**AR1:**

**We agree with the referee that, the Doppler velocity may offer a special insight into the distinction between different microphysical processes. We didn't use the Doppler velocity from vertical scans in this paper since the spatial and temporal shifts between the QVPs and vertical scans can dim the robustness of the results.**

**The QVPs shown in this paper were calculated at an elevation of 18$^{\circ}$ . It is thus not convincing to apply directly the Doppler measurements of radar vertical scans to distinguish aggregation and riming in this paper, considering the temporal and spatial offsets between QVPs and vertical scans.**

**We thus rephrased Section 5 and discussed the possibility of using Doppler velocity at vertical scans to complement the analysis in the paper.**

**P26, line 11-20: "…*While there were clear signs of other processes like riming and aggregation, a distinction between these two processes is still difficult with the available observations. Doppler velocities at the vertical pointing mode were analyzed but the observed values (between 1 - 2 m/s) still makes the distinction ambiguous. Furthermore, the exact time from the QVP and vertical pointing scans cannot be matched, and one has to be careful when comparing the QVP with vertical scans. Additional analysis in conjunction with other independent observations e.g. from microwave radiometers, lidars and cloud radars which were deployed at the JOYCE site is also required for a better distinction between riming and aggregation, which is the focus of an ongoing study. …*"**

RC2: Polarimetric measurements from the three radars are used in the microphysical analysis, as suggested by the title of the paper. Additionally, ground based instruments are used, not only to confirm, but also or to complete this analysis. The title should suggest the use of such complement.

**AR2:**

**Thanks for the suggestion, the title of the paper was revised to "Precipitation and Microphysical Processes Observed by Three Polarimetric X-Band Radars and Ground-Based Instrumentation during HOPE" in order to complete the use of disdrometers/rain gauges/MRRs for the microphysical analysis.**

SPECIFIC COMMENTS:

RC3: Except for rain rate estimation, rain (or hail or melting snow) attenuation impacting X-band measurements is not mentioned at all in the paper. Indeed, the analyses use Z and ZDR, potentially biased by such attenuation. Are Z and ZDR corrected for attenuation?

**AR3:**

**We didn't perform attenuation correction in this paper. Firstly, the precipitation during HOPE is not intense and the HOPE site is close to the KiXPol and JuXPol, within 10 km. Thus, for the low rain rate, the attenuation effects due to precipitation can be negligible. Secondly, for rain rate > 8 mm/h (the duration of rain rate > 8 mm/h during HOPE is only around 1 hour), R-Kdp relation which is unaffected by attenuation effects is employed instead of R-Z relation.**

**To make it clearer, the following statement is added: (p11, lines 24-26) "…*Z and $Z_{DR}$ attenuation along each radial is neglected since the rain intensities were generally low over the HOPE area*.…". (see also response to Reviewer#2, AR1)**

RC4: p7 lines7-9: Why are BoXPol disdrometer measurements not used, while MRR observations at the same site are?

**AR4:**

**Sorry for the confusion. The disdrometer measurements close BoXPol are not used for the precipitation analysis presented in Section 3, since BoXPol disdrometers are ~50 km away. Thus considering the spatial and temporal variability of precipitation, the statistical analysis of precipitation over the central HOPE area shown in Section 3 is only performed with the disdrometers/rain gauges in the vicinity of Juelich. In Section 4, the microphysical processes are analyzed with a combination of polarimetric radars and ground-based instruments. The MRR data in Section 4 is only used for analyzing microphysical process (size sorting in Fig.11) close to BoXPol.**

**To make this clearer, the sentence was modified. "…*Disdrometer observations at***

*BoXPol which is ~48.5 km away from JuXPol are not taken into account in Section 3 when statistically analyzing the precipitation over HOPE, considering the spatial and temporal variability of rainfall…."* (p8 lines3-5).

RC5: p9 lines 15-16: I do not understand how the distribution shown in Fig.4 results from individual measurements of disdrometers instead of averaging over disdrometer sites at a single time step. Does that mean, for instance, that 80 hours of [0,1] mm/h rain have been obtained by summing rainfall observation time over the N disdrometers (thus representing 80/N hours each)?

**AR5:**

**Sorry for the confusion, we revised the sentence in p9 lines 15-16.**

**(now P9 lines 22-23) "…*The distribution of rain intensities was calculated based on individual measurements of disdrometers over the HOPE area...*"**

**Figure 4 was calculated following the steps below:**

**(1) for each disdrometer, the distribution of rain intensities was calculated individually.**

**(2) averaging the rain hours over the number of disdrometers at each rain rate intervals. For instance, at [0,1] mm/h rain rate, when one disdrometer has a duration of 100 hours and another one has 60 hours, an average over the two disdrometers is 80 hours of [0,1] mm/h, as shown in Figure 4.**

RC6: p12 lines 16-17 and p13 line 13: the lack of consistency between rain rates also probably suffers from the representativeness error impacting BoXPol measurements (higher altitude of sampling volume).

**AR6:**

**Thanks to the referee's suggestion. We revised the sentence to better explain the low rain rate from BoXPol measurements and added "the high altitude of sampling volume" as one of the error sources which result in a low rain rate of BoXPol.**

*"…JuXPol and KiXPol are in a better agreement with the surface measurements*

*than BoXPol for the very low rain rates, which probably suffers from the effects of*

*non-uniform beam filling effects due to the much larger distance from the HOPE*

*area (Giangrande and Ryzhkov, 2008) and higher altitude of sampling volume of*

*BoXPol…." (p12, lines 14-18)*

*"…again probably caused by beam broadening (Giangrande and Ryzhkov, 2008)*

*and high altitude of sampling volume of BoXPol over the HOPE area…." (p13, line*

*13-14)*

RC7: p14 Fig.7: When comparing KixPol and JuxPol rain accumulation, the south-west quarter of the panels shows significant differences. How could this be explained? Is this a problem of projection on ground?

**AR7:**

**The discrepancies in the south-west quarter is not caused by the projection on the**

**ground. It could be because:**

**(1) due to the beam blockage of KiXPol, the radar signals in the south-west**

**direction are missing. KiXPol was deployed at an altitude ~116 m, while close to**

**KiXPol, where the significant discrepancies occur, the altitude is above 200 m.**

**(2) JuXPol and KiXPol didn't measure at the same time and the same location. The**

**time and space shifts exist between JuXPol and KiXPol and result in the different**

**precipitation patterns especially when it is close to the radar. JuXPol is roughly 10**

**km away. At an elevation of 4,5° it reaches a height of roughly 800 m above KiXPol.**

**The time differences between the two radar measurements are in an interval of 5**

**min, which should be also taken into account. We thus proposed the**

**reconstruction of the three radar observations needs to be done for better**

**comparisons.**

**To make this clear, we added the explanation in the text. "…*A 30-min rain***

***accumulation over the inner HOPE area on 29 May 2013 shows that, the three***

***radar estimates result in an overall agreement of the rough precipitation pattern***

***(Figure 7). However, when we zoomed into details and noticed also the minor***

***differences between these patterns, e.g., lower precipitation observed by BoXPol***

***and missing pixels near KiXPol and JuXPol. Bins close to KiXPol and JuXPol were***

***contaminated by ground clutters while the beam broadening and height at the***

***larger ranges deteriorates the similarity between the BoXPol and KiXPol/JuXPol***

***estimates (Fig. 7). The different radar observation scenarios, i.e., at an elevation of***

***4.5° JuXPol reaches 750 m above KiXPol and the time differences between the two***

*radar measurements are up to 5 min, also needs to be considered. A combination of the three radar observations will definitely be an advantage to reconstruct the precipitation patterns over the HOPE area in a future study…."*

RC8: p25 line 3: About measurements indicative associate updraft, what about radar Doppler measurements at vertical incidence? (see general comment)

**AR8:**

**we agree with the referee that, the Doppler velocity may offer a special insight into the distinction between different microphysical processes, as mentioned above. We thus revised Section 5 and discussed the possibility of using Doppler velocity at vertical scans to complement the analysis in the paper. (See response to Reviewer#1, AR1)**

TECHNICAL CORRECTIONS:

**AR:**

**Thanks to the careful reading of the referee, we corrected all the technical errors listed below according to the suggestions.**

RC9: p1 line 21: replace "another three disdrometers" by "three other disdrometers".
**AR9: Corrected. (now p1 line 22)**

RC10: p14 line 9: remove "s" in radars (replace "three radars estimates" by "three radar estimates").
**AR10: Corrected. (now p14 line 11)**

RC11: p17 line 12: replace "30 dBz" by "30 dBZ".
**AR11: Corrected. (now p18 line 8)**

RC12: p23 line 11: add "of" after "a region"

**AR12: Corrected. (now p23 line 20)**

[revised manuscript text omitted]

---

## Author Comment (AC2) · 6 May 2016

**Response to Anonymous Referee #2**

**We thank the reviewer for his/her valuable comments. We responded to the comments and made modifications in the paper accordingly.**

**RC: reviewer comments**

**AR: author response**

General Comments

RC1: This paper presents some data from a two months experiment. The examined dataset included three polarimetric X-band weather radars supplemented by MRRs, disdrometers and rain-gauges. The focus of the paper is on the radar observations. The recorded rain events were of low intensity and this didn't permit a more advanced evaluation of radars performance. Thus, instead of just showing some daily statistics and example data from three rain events with typical stratiform rain characteristics, the authors could present methods of data processing. For example, they have a network of three radars which overlap in the area of interest and, thus, a detailed comparison between the radars (and the rest of sensors like the MMR, disdrometers and raingauges) could be performed. Furthermore, a method for construction of a mosaic with the quality controlled measurements from the three radars would be meaningful as a first data analysis. Also, the authors don't even mention the basic and critical processing algorithms of the radar data like the attenuation correction scheme and the handling of melting layer (bright band) effect on the estimated rain field.

**AR1:**
**Thanks for the comments. We started the paper with the description of the HOPE experiment and associated instrumental set up, followed by an analysis of the**

**three case studies from three polarimetric X-band radar observations. The paper**

**focuses on multi measurement capability to improve or assess microphysical**

**process knowledge of precipitation evolution. We thus present here the ability of**

**three radars for combined observations of precipitation.**

**The HOPE campaign was aimed at an assessment and improvement of the high**

**resolution climate model ICON (for details about ICON, see**

**http://www.mpimet.mpg.de/en/communication/news/focus-on-overview/icon-de**

**velopment/) with the available observations, as we stated in the introduction, and**

**our results presented here approach to the aim of the campaign.**

**About the construction of a mosaic with the quality controlled measurements from**

**the three radars, a paper from Mauro et al., which is focusing on the radar**

**composite for HOPE is close to submit. We thus didn't repeat the study in our**

**paper here.**

**We didn't perform attenuation correction in this paper. Firstly, the precipitation**

**during HOPE is not intense and the HOPE site is close to the KiXPol and JuXPol,**

**within 10 km. Thus, for the low rain rate, the attenuation effects due to**

**precipitation can be negligible. Secondly, for rain rate > 8 mm/h (the duration of**

**rain rate > 8 mm/h during HOPE is only around 1 hour), the R-Kdp relation which is**

**unaffected by attenuation effects is employed instead of R-Z relation. Melting layer**

**effects can also be neglected for rainfall attenuation at least for JuXPol and KiXPol**

**because of their close proximity to the site. For BoXPol we use the 1° elevation and**

**over the HOPE area the radar beam height is at ~860 m height, which is below the**

**melting layer according to radiosonde observations during the precipitation**

**duration. To make it clearer, we stated this in the paper. (p11, lines 24-26) "…*Z and***

***$Z_{DR}$ attenuation along each radial is neglected since the rain intensities were***

***generally low over the HOPE area*.…"**

Specific Comments

RC2: Section 2.1, Fig. 1: The setup of the systems shown in Fig. 1 is not optimal at all.

Most of the systems (including two radars) are within 5 km distance. If this setup was intended for e.g. the study of small scale spatial distribution of rain this was shown in the paper.

**AR2:**

**Thanks for the comments. We agreed with the reviewer that the setup of the**

**systems was not optimized for precipitation observations. However, our influence**

**on the setup was limited and the campaign was especially designed for cloud**

**process observations and only to a lesser extent for precipitation observations.**

**However, we only concentrated on the observations from the precipitation**

**monitoring instruments over the HOPE area, as we stated on Pg 2 line 13-15.**

**All the systems were deployed within 10 km distance and used to verify and**

**improve the high resolution climate and weather forecast model ICON over the**

**HOPE area ( for details about ICON, see**

**http://www.mpimet.mpg.de/en/communication/news/focus-on-overview/icon-de**

**velopment/). The results presented in this paper will be useful to evaluate and**

**improve the ICON model and a paper, which evaluates the cloud and precipitation**

**performance of the ICON model with available measurements, is in preparation.**

RC3: p. 10, Fig. 4: The daily accumulated precipitation from the 7 disdrometers in Fig.

4b has larger range (minimum, maximum) compared to the range from the 3 rain gauges and the 7 disdrometers in Fig. 4a in some days (e.g. on 26 April), while obviously it should be less.

**AR3:**

**We do believe that the reviewer discussed on Figure 3 since Figure 4 has only one**

**panel. We are sorry for the confusion. To make this clearer, we revised Figure 3 and**

**used different colors for rain accumulation and duration.**

**Figure 3a shows the daily rainfall accumulation with the range of bars indicating**

**the range of rain accumulation (mm), while Figure 3b shows the precipitation**

**duration and the range of bars is the precipitation duration in hours.**

RC4: p. 14, lines 8-14: The conclusions of the authors about Fig. 7 are contradictory.

First they say that precipitation patterns observed by the three radars, but immediately after the mention a lot of the many reasons why the observed patterns are different (which is the correct conclusion). They propose that a reconstruction of the precipitation pattern using a combination of all the radar data should be made, but as it was noted in the general comments they don't try to implement such a method.

**AR4:**

**Sorry for the confusion. For clarification, we rephrased the text (Pg 14, line, 12-21).**

**First, we discussed the overall agreement of the rough precipitation patterns**

**observed by the three radars and second, we zoomed into details and noticed also**

**the minor differences between these patterns, e.g., lower precipitation observed**

**by BoXPol located far away from the other two radars. We rephrased the sentences**

**and explained possible reasons responsible for the differences. To eliminate these**

**discrepancies, we thus proposed to make a reconstruction with the three radars in**

**a future study (a paper on the three-radar reconstructed precipitation is about to**

**be submitted).**

**(Pg 14, line, 10-23) "...*A 30-min rain accumulation over the inner HOPE area on 29***

***May 2013 shows that, the three radar estimates result in an overall agreement of***

***the rough precipitation patterns. However, when we zoomed into details and***

***noticed also the minor differences between these patterns, e.g., lower precipitation***

***observed by BoXPol and missing pixels near KiXPol and JuXPol. Bins close to KiXPol***

***and JuXPol were contaminated by ground clutters while the beam broadening and***

***height at the larger ranges deteriorates the similarity between the BoXPol and***

***KiXPol/JuXPol estimates (Fig. 7). A combination of the three radar observations will***

***definitely be an advantage to reconstruct the precipitation patterns over the HOPE***

***area in a future study*. *The different radar observation scenarios, i.e., at an***

***elevation of 4.5° JuXPol reaches 750 m above KiXPol and the time differences***

*between the two radar measurements are up to 5 min, also needs to be considered.*

*Since no adjustments of the R-$Z_H$ and R-$K_{DP}$ relations were made, these results are*

*very promising. The three radar estimates together with the direct comparisons*

*with the rain gauges and disdrometers allow to attribute robust error estimates to*

*these precipitation fields, which will be very valuable when compared with model*

*simulations.*"

RC5: p. 16, Fig. 9: There are not evident melting layer characteristics in the RHIs, even though it is mentioned in the text to move from 2100 m height down to 830 m during the event. It would be useful to include in Table 1 (or in a separate table) the operational parameters of the radar (like beamwidth, antenna rotation rate, sampling frequency etc.)

**AR5:**

**We agreed with the reviewer that there is no evident melting layer visible in RHIs.**

**However, the radionsondes launched at 11 UTC, 13 UTC, 16 UTC, and 23 UTC were**

**able to capture well the descent of 0°C level: from 2100 m to 830 m height, as**

**stated in Pg. 15 lines 13-16.**

**We added the parameters of the radars in Table 1. We also mentioned in the paper**

**that the operational parameters of JuXPol and BoXPol can be found in Diederich et**

**al. (2015a) and for KiXPol under www.imk-tro.kit.edu/english/5438.php.(Pg 4 lines**

**12-15).**

RC6: Section 4.2: In this section some data from MRR and disdrometers are shown.

As it was noted in the general comments the authors probably have enough data from the radars and these sensors to make a more detailed and useful comparison of their measurements. For example, a comparison of radar RHI data over (or near) the

MRR site and MRR data would be an interesting comparison and study of the melting layer characteristics.

**AR6:**

**We agree with the reviewer that a comparison between MRR and BoXPol would be**

**interesting. However, MRR is in a distance of 200 m away from BoXPol (BoXPol RHI**

**scan every 5 min) and it is noticed that the precipitating system was passing by the**

**MRR within 10 min (Figure 12), i.e., two RHI scans from BoXPol. The coarse**

**temporal resolution of BoXPol makes it difficult to compare directly the MRR and**

**BoXPol observations over the MRR site.**

**To make this clearer, we added sentences in the revised paper (Pg 21, lines 13-15)**

**"...*However, the coarse temporal resolution of BoXPol RHI scans (every 5 min)***

***makes it difficult to compare directly the MRR observations with BoXPol over the***

***MRR site....*"**

RC7: p. 21, lines 22-23: Why consider MRR data at 600m height as a reference (and not e.g. rain gauge data) and conclude that the Parsivels are overestimating rainfall rate? The MRR should be reduced to ground level using the time delay due to the average fall velocity of the droplets to have a proper comparison.

**AR7:**

**We agree with the reviewer that a better comparison can be conducted between**

**the surface precipitation measurements. However, the near surface data of MRR**

**can't be used since MRR derives extremely high rain rates which can not be**

**trustable. We stated in Pg 9 lines 3-4, "...*due to the near field scattering effects,***

***MRR observations at the first three gates are not used....*" Therefore, we only use**

**MRR data at 600 m height in Figure 12f.**

**Also, considering the effects from size sorting and other possible microphysical**

**processes, the rain rate at higher levels is usually higher than on the ground (e.g.,**

**rain depleted by evaporation). Also, the two Thies disdrometers close to the**

**Parsivel show a relatively lower maximum rain rate on the ground, too. We thus**

**conclude that the Parsivel is overestimating the rain rate. To make this clearer, we**

**rephrased the text in the revised paper. (Pg 22, lines 1-5) "...*Considering the effects***

*from size sorting and other possible microphysical processes, the rain rate at high altitudes is usually higher than on the surface. The two Thies disdrometers close to the Parsivel, which provide measurements at 1 min time intervals, also show a smaller maximum rain rate near the ground and corroborate these findings. We thus conclude that the Parsivel is overestimating the rain rate..."*

RC8: p. 23, Fig. 13: A comparison of QVPs and data from RHIs would be useful to understand the difference of QVP from actual vertical profiles and the limitations of this method.

**AR8: We agree with the reviewer that a comparison between QVPs and RHIs is useful. However, going into details of each method used in a publication is beyond the scope of a new publication applying different methods. A detailed description of QVPs can be found in Ryzhkov et al.(2016) and Troemel et al. (2014a). Ryzhkov et al. (2016) also included a RHI-QVP comparison. Ryzhkov et al. (2016) discussed in detail the benefits of QVPs and their superiority compared to RHIs with respect to the detection of microphysical processes. We thus did not discuss in detail the limitations of the QVP method in this paper. Additionally, the RHI measurements from JuXPol are available in 5 min and only restricted to a narrow azimuthal direction. Consequently, we decided to show QVPs only in Section 4.3.**

[revised manuscript text omitted]